# LongShield: Scalable Distributed Differentially Private Training for Long-Context LLMs

## Abstract

Large language models excel at in-context learning, but can memorize sensitive sequences and enable membership-inference and extraction attacks. Differential privacy (DP) offers provable protection, yet DP training remains costly for long contexts: prior work primarily targets short-sequence DP fine-tuning, and the strongest public DP pretraining scales only to 1B parameters with 1,024 tokens contexts.

We focus on providing DP guarantees for long data. However, the state-of-the-art DP solution ZeRO-DP is optimized for small sequences and fails to scale to long sequences due to the single-GPU memory ceiling for the unsharded activations under FSDP. Moreover, straightforward context extension techniques like CP do not work out of the box with ZeRO-DP, as the ghost norm overhead dominates compute and communication for long sequences.

We introduce LongShield, a memory- and communication-efficient context-parallel DP training method that closes the performance gap to non-DP while enabling long-context scaling on modest GPU budgets. LongShield keeps per-sample gradients shards local to each GPU to avoid full materialization, overlaps per-sample gradient aggregation with backward computation to sustain throughput, and enables DP-safe activation checkpointing to extend context further. These system changes leave the underlying DP algorithm and accounting unchanged, and use flat clipping for best convergence. On Llama 3.1 8B with 4× NVIDIA H100 GPUs, LongShield scales sequence length from 4k to 16k compared to the state-of-the-art ZeRO-DP, achieves linear sequence-length scaling, shrinks the throughput gap from 67% to 8.9% while matching non-DP memory usage, and reaches a 64k context length with activation checkpointing. These results show that long-context DP training is practical on modest GPU budgets.

## 1 Introduction

Modern LLMs support and benefit from increasingly large context lengths (Jacobs et al., 2023; Kryściński et al., 2022; Huang et al., 2024). For example, Llama 3.1 (Grattafiori et al., 2024) and Qwen 2.5-1M (Yang et al., 2025) support sequence lengths of 128k and 1M, respectively. Advances in LLMs' long-context capabilities depend heavily on long, high-quality datasets, such as full patient records and proprietary codebases, which contain sensitive information.

Although LLMs excel at in-context learning, they can memorize rare sequences (Carlini et al., 2022; Nasr et al., 2023), making the model vulnerable to membership-inference attacks (MIA) (Shokri et al., 2017). Differential privacy (DP)[1] techniques are the gold standard for provably constraining privacy leakage from the underlying training data and offer formal protection against such memorization (Abadi et al., 2016; VaultGemma Team, 2025).

However, DP training remains costly, especially with longer contexts. Early academic results (Li et al., 2021; Bu et al., 2023d) focused on DP fine-tuning over tiny context lengths (around 100 tokens for table-to-text generation on the E2E dataset (Novikova et al., 2017)); the strongest public

---

[1]DP is the abbreviation for data parallelism in distributed training literature. In the context of our paper, DP always means Differential Privacy, following existing literature. We will explicitly say "data parallelism" if encountered.

DP pretraining (VaultGemma Team, 2025) to date targets 1B-parameter models with 1024-token contexts, which is orders of magnitude smaller than today's non-DP models with up to 1M-token contexts (Yang et al., 2025).

This is because state-of-the-art DP-SGD (Abadi et al., 2016) (or its variants) requires large effective batch sizes to control noise, which competes for memory capacity with long contexts. Practitioners therefore shorten the context to fit more batches (VaultGemma Team, 2025), sacrificing the long-context capability modern applications require (Liu et al., 2023b; Bai et al., 2023; Grattafiori et al., 2024; Yang et al., 2025). Moreover, using shorter context lengths increases the token throughput, which is crucial for offsetting the non-trivial slowdown introduced by DP.

Nevertheless, it is critical to enable long-context capability for LLMs under DP to protect long private data where existing DP solutions fall short. In the non-private domain, context extension continued pretraining (CPT) is the standard approach to enable long-context capability for models pretrained on small sequences (Grattafiori et al., 2024; Yang et al., 2025; Fu et al., 2024; Xiong et al., 2023). However, CPT has very different requirements than the from-scratch pretraining approach taken by DP work to date (see Table 1), and existing SOTA distributed DP solutions like ZeRO-DP cannot be used with.

This is because, even with infinite GPU resources, long contexts do not fit given the ZeRO-DP sharding layout — chosen to maximize throughput — due to the single-GPU memory ceiling for the unsharded activations under FSDP (See Section 2). Moreover, straightforward context extension techniques, such as context parallelism (CP) (Liu et al., 2023a), do not work out of the box with ZeRO-DP, as the resulting ghost overhead demands $O(T^2)$ compute and $O(T)$ communication, which is expensive with long contexts.

In this paper, we show how CPT can be adapted to the DP setting. A key observation is that context extension requires orders of magnitude fewer tokens than pre-training from scratch — just 0.5B to 5B tokens (Fu et al., 2024) — meaning that throughput requirements are much lower. Lower throughput pressure means that the microbatch size (MBS) can be made much smaller, leaving memory capacity for the sequence dimension T. Instead of using MBS to control noise, we compensate with more gradient accumulation steps, effectively controlling noise by setting the global batch size (GBS).

|  | TPS requirements | MBS | T |
|---|---|---|---|
| DP pretraining from scratch | high | large | small |
| DP context extension CPT | low | small | large |

Table 1: Performance requirements between DP PT from scratch and DP context extension CPT. TPS = tokens per second; MBS = microbatch size; T = sequence length.

**Key insights:** We adopt the pure gradient-sample (GS) approach to avoid ghost overhead. However, challenges remain in tackling the heavy memory pressure of saving per-sample gradients across the entire model. In contrast to SOTA ZeRO-DP, which avoids tracking per-sample gradients over the whole network using ghost clipping, we realize a unique sharding opportunity that is otherwise unavailable under FSDP to provide memory scalability that is crucial for larger models (a larger model corresponds to a larger per-sample gradient overhead).

The sharding opportunity comes at the cost of additional communication. We analyze the trade-off between output-stationary and input-stationary communication patterns. We choose the input-stationary pattern for better scalability and hide the communication with independent computation.

**Contributions:** To enable scalable and efficient long-context distributed DP training that satisfies the requirement of context extension CPT under DP, we introduce LONGSHIELD: a DP training recipe that scales context, not cost. We make the following contributions:

- We adopt a pure GS approach to avoid ghost overhead at long contexts and integrate it with context scaling methods, such as CP, to achieve linear sequence scaling. We treat this as a baseline. It beats SOTA ZeRO-DP in terms of achieved sequence length and the throughput. However, it still suffers from the standard memory penalty of storing the per-sample gradients for the entire model.

- We further reduce per-GPU DP memory overhead by sharding per-sample gradients within the context-parallel domain. Such an opportunity does not exist in prior SOTA distributed solutions DP-Zero (Bu et al., 2023a). We minimize the required communication and hide it using independent computation to avoid throughput reduction.

- We use activation checkpointing (Chen et al., 2016), which is incompatible with prior DP frameworks (Yousefpour et al., 2021; Li et al., 2021; Bu et al., 2023d;a). This enables additional context scaling under limited resources at the cost of an extra forward pass, resulting in approximately a 33% reduction in throughput.

LONGSHIELD achieves 4× context scaling compared to ZeRO-DP Bu et al. (2023a) under 4× H100 GPUs under various LLAMA 3 family of models. DP-aware activation checkpointing provides up to 4× additional context scaling. Meanwhile, we significantly close the throughput gap between non-private baselines compared with prior SOTA ZeRO-DP (e.g., 67% to 8.9% on LLAMA 3.1 8B), while maintaining non-DP memory usage. Preliminary and related work is included in appendix Section A.

## 2 CHALLENGES OF LONG-CONTEXT DP

The state-of-the-art distributed DP solution ZeRO-DP Bu et al. (2023a) leverages the zero redundancy optimizer (ZeRO) (Rajbhandari et al., 2020) or fully sharded data parallelism (FSDP) (Zhao et al., 2023; PyTorch Documentation, 2025) to scale the SOTA single-GPU efficient DP methods (Bu et al., 2022; 2023d). However, ZeRO-DP (Bu et al., 2023a) cannot help sequence scaling and is therefore not suitable for the context extension CPT task under DP. ZeRO-DP gets out of memory error (OOM) even with infinite H100 GPUs for sequence length 32k, 16k, and 8k for LLAMA 3.2 1B, LLAMA 3.2 3B, and LLAMA 3.1 8B, respectively. Below, we first explain why ZeRO-DP fails to scale, and then discuss suitable context scaling techniques LONGSHIELD use to scale.

Under FSDP, each GPU holds a sharded model state $O(M/N)$ (M for model state space and N for number of GPUs) and unsharded activations $O(MBS \times T \times L \times h)$ (MBS for micro-batch size, T for sequence length, L for number of layers, and h for hidden size). The sum of the sharded states and the unsharded activation needs to be smaller than the GPU physical memory size.

FSDP is subject to a hard memory ceiling for sequence length scaling. Even if N reaches infinity, and the model states space reaches zero. The unsharded activation must fit within the single-GPU memory limit. Even choosing MBS=1, there's an upper limit for sequence length under FSDP.

Table 2 shows the maximum achievable sequence length (power of two) across various MBS for LLAMA 3.2 1B, LLAMA 3.2 3B, and LLAMA 3.1 8B over 1, 4, and infinite H100 (80GB) GPUs. The benefits are marginal beyond 4 GPUs.

| | 1B | | | 3B | | | 8B | | |
|---|---|---|---|---|---|---|---|---|---|
| Num GPUs | 1 | 4 | $\infty$ | 1 | 4 | $\infty$ | 1 | 4 | $\infty$ |
| MBS=1 | 16384 | 16384 | 16384 | 4096 | 8192 | 8192 | OOM | 4096 | 4096 |
| MBS=2 | 8192 | 8192 | 8192 | 2048 | 4096 | 4096 | OOM | 2048 | 2048 |
| MBS=4 | 4096 | 4096 | 4096 | 1024 | 2048 | 2048 | OOM | 1024 | 1024 |
| MBS=8 | 2048 | 2048 | 2048 | 512 | 1024 | 1024 | OOM | 512 | 512 |

Table 2: The maximum achieved sequence length (power of 2) under various MBS for LLAMA 3.2 1B, LLAMA 3.2 3B, and LLAMA 3.1 8B over 1, 4, and infinite H100 (80GB) GPUs.

In contrast to FSDP, two types of approaches help with sequence scaling: (1) context parallelism (CP) implements a spatial version of the flash attention (FA) (Dao et al., 2022; Dao, 2024), which enables linear scaling with respect to the number of GPUs, despite slower inter-node communication and quadratic attention costs; (2) memory optimization techniques like activation checkpointing recompute activations during backward to avoid saving all activation tensors. However, these techniques do NOT work directly with ZeRO-DP or will encounter significant system overhead.

Naive combination of ZeRO-DP with CP introduces significant ghost overhead. The mixed ghost norm heuristic (Bu et al., 2022) prefers to use ghost clipping, especially for large layers like the

final linear layer. For example, mixed ghost norm choose ghost for LLAMA 3.1 8B final linear layer up to T= 16k. However, the ghost norm is 4× more FLOPs than directly evaluating the per-sample gradient, and the final dot product between two large intermediate tensors ($O(BT^2)$) causes a similar delay, according to our profiling, due to the reduction nature. This causes 8× slowdown for the largest layer. Communication is more of an issue, as input tensors to the ghost norm are context-distributed, requiring an all-gather (AG) or ring-exchange that is similar to ring attention (Liu et al., 2023a). Instead, LONGSHIELD builds with a GS approach to avoid ghost overhead at long context.

Meanwhile, ZeRO-DP (Bu et al., 2023a) or any hook-based DP approach (Yousefpour et al., 2021; Li et al., 2021; Bu et al., 2023d;a) is not compatible with AC. The forward hooks used in DP frameworks attempt to capture activation so that the backward hook can evaluate the per-sample gradient norm using either the gradient sample or the ghost clipping method. However, this hook will capture both the activation that is supposed to be released and the recomputed activation, resulting in incorrect behavior. We show proper hook management in Section 3.3.

# 3 LONGSHIELD DESIGN

As discussed in Section 2, directly extending SOTA ZeRO-DP to longer contexts faces two challenges: (i) the ghost norm overhead at long distributed contexts, and (ii) incompatibility with memory optimization techniques like activation checkpointing (Chen et al., 2016).

LONGSHIELD therefore avoids ghost clipping and instead adopts the pure grad sample (GS) approach. However, scaling GS with context parallelism faces new challenges. The GS method is notorious for preserving the per-sample gradient over the entire model, adding significant memory pressure and limiting scalability, especially for large models. ZeRO-DP switches to the ghost norm to prevent this, but the ghost norm has quadratic complexity in sequence length, which is a cost that we cannot afford for context scaling.

Instead, LONGSHIELD takes advantage of a new sharding opportunity that is otherwise unavailable in ZeRO-DP with FSDP. We introduce CP per-sample gradient sharding in Section 3.1.

However, CP per-sample gradient sharding is not free and introduces additional communication challenges compared with FSDP. Section 3.2 analyzes the tradeoff between the *output-stationary* and the *input-stationary* communication patterns, and discusses their communication volume and how to avoid the throughput penalty resulting from exposed communication.

Finally, we introduce DP-compatible activation checkpointing in Section 3.3, which enables additional context scaling capability.

## 3.1 DP MEMORY–COMMUNICATION TRADE-OFF BETWEEN CONTEXT PARALLELISM AND FSDP

Let's consider a generalized linear layer with dimensions $p$ by $d$, and evaluate its per-sample gradient of shape (MBS, $p$, $d$). Figure 1 illustrates the initial sharding state with a toy 2-GPU example, comparing FSDP and CP.

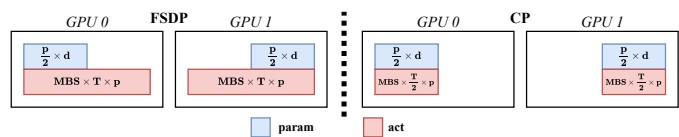

Figure 1: Activation and parameter sharding under FSDP and CP

FSDP shards the model states and enables batch scaling where GBS = 2 × MBS. However, the single-GPU activation of the shape (MBS, T, $p$) is not sharded and therefore has limited sequence scaling capability.

When it comes to CP, one can also use a distributed model state, where each GPU holds only a shard of the parameters. Instead, the activation of shape (MBS, T, $p$) is sharded over the sequence length dimension, and each GPU holds an activation of shape (MBS, T/2, $p$).

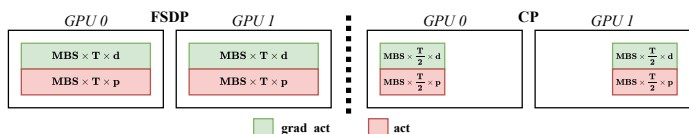

Figure 2: Activation and activation gradient sharding under FSDP and CP

Figure 2 shows the sharding states before computing the per-sample gradients, where the upstream activation gradients are ready and follow the corresponding FSDP and CP sharding strategy. Notice that in FSDP, the entire sequence length T is local to each GPU, and the per-sample gradient can be evaluated locally without any communication. Evaluating per-sample gradient requires the following einsum operation (we use B for MBS):

$$\text{per-sample grad} = \text{einsum}(BTp, BTd \rightarrow Bpd, \text{ act}, \text{ grad\_act}).$$

However, local computation for the per-sample gradient under CP only yields partial results, which means that some forms of communication among context-parallel GPUs are required. We provide a careful analysis of different communication patterns and their consequences in Section 3.2.

On the other hand, FSDP incurs the full memory overhead of saving the per-sample gradient, as shown in Figure 3. The per-sample gradients on each GPU are not shardable as they correspond to different samples. However, per-sample gradients can be sharded within the CP domain, as they correspond to the same MBS, which is significantly more scalable compared to FSDP.

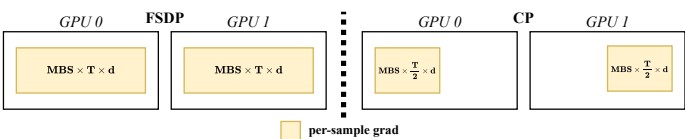

Figure 3: Per-sample gradient under FSDP and CP

### 3.2 COMMUNICATION FOR PER-SAMPLE GRADIENT WITH CONTEXT PARALLELISM

Evaluating per-sample gradient under CP (from Figure 2 to Figure 3) can be achieved with multiple approaches, with distinct communication patterns and bandwidth requirements.

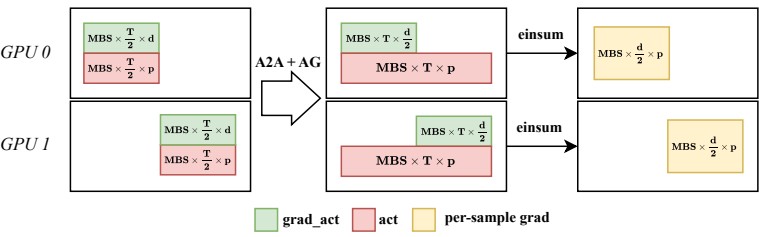

Figure 4: Output Stationary DP under CP

**Output-stationary pattern.** An output-stationary approach (Figure 4) exchanges the input tensor (i.e., activation and activation gradients) followed by the per-sample gradient einsum operation. For example, we can all-to-all (A2A) exchange the activation tensor (shape changing from (MBS, T/2, p) to (MBS, T, p)) and then all-gather (AG) the activation gradient tensor (shape transferred from (MBS, T/2, d) to (MBS, T, d/2)). Performing the per-sample grad einsum operation then directly yields a complete shard of per-sample gradient (shape (MBS, d/2, p)). One optimization is to apply A2A on the large tensor and AG on the smaller tensor (by comparing $d$ and $p$) to reduce communication

volume. This effectively makes the sharding dimension configurable (one can choose either from $d$ or $p$). One needs to align other model state sharding such that the optimizer step does not require additional communication.

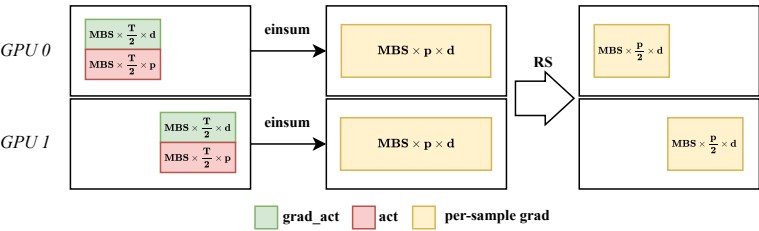

Figure 5: Input Stationary DP under CP

**Input-stationary pattern.** An input-stationary approach (Figure 5) computes per-sample gradients based on local activation and activation gradient shards, followed by exchanging the output per-sample gradient tensor instead. Directly evaluating per-sample gradients based on local activation and activation gradient shards yields a partial per-sample gradient with shape (MBS, p, d). Then, a reduce-scatter (RS) operation aggregates and produces the complete per-sample gradient shard of shape (MBS, p/2, d).

For long contexts, we prefer the input-stationary approach, as the required communication does not scale with sequence length. AG (even for the smaller tensor between activation and activation gradient) can be a performance bottleneck over a long context with large T. Meanwhile, the output-stationary approach is hard to overlap communication with computation in practice. At the time we enter the backward hook, the data gradient operation is completed, and there is no independent computation to overlap. This causes the input tensor exchange to be on the critical path, exposing the communication latency and resulting in lower throughput. However, for the input-stationary approach, one can post the RS with the following independent computation: backward data gradient computation of the previous layer.

### 3.3 DP-compatible Activation Checkpointing

Activation checkpointing (AC) (Chen et al., 2016) is a popular memory optimization technique that avoids saving full activations during the forward pass and recomputes the required activations during the backward pass. However, this approach is not compatible with mainstream DP frameworks (Yousefpour et al., 2021; Li et al., 2021; Bu et al., 2023d;a) due to **incorrect** activation memory management. Mainstream DP frameworks (Yousefpour et al., 2021; Li et al., 2021; Bu et al., 2023d;a) adopt the hook-based method from Opacus (Yousefpour et al., 2021) to track and release activation that will be used for per-sample gradient norm evaluation.

All trainable modules will be wrapped with DP forward and backward hooks. The DP forward hook captures the reference to the input activation of the module. The DP backward hook captures the reference to the activation gradient tensor, uses the tracked activation from the forward hook to evaluate per-sample gradients, and frees the tracked activation.

Direct integration with AC fails with hook-based DP methods. During the forward pass, the module under the AC regime is supposed to release the activation and recompute during the backward pass. However, the forward hook captures a reference to activation and therefore prevents the release of activation, which defeats the original purpose of activation checkpointing. Moreover, the forward hook will fire again during the backward pass when forward recomputation is triggered. This captures the references to the newly recomputed activation. The backward hook evaluates per-sample gradient using the new activation and frees it, leaving the activation captured by the initial forward pass dangling, causing the training to go OOM after some steps eventually.

To preserve the intended AC behavior under DP, we disable forward hooks under the AC region to prevent tracking references to activation that is supposed to be freed. Any trainable module outside the AC region is left untouched. After the forward pass and prior to the backpropagation, we re-enable

forward hooks for all DP modules under the AC region such that the recomputed activations can be captured and the backward hook can use the captured activation to compute the per-sample gradient.

## 4 EVALUATION

### 4.1 METHODS

Our experiments were conducted on a node with four H100 80 GB GPUs and 900 GB/s NVLink. We build LONGSHIELD on top of the SOTA DP framework Opacus (Yousefpour et al., 2021) and leverage TorchTitan (Liang et al., 2025) distributed training support on LLMs. We use TorchTitan's internal tools to report throughput (in TPS = tokens per second per GPU) and monitor peak memory usage. We evaluate three popular variants of the LLAMA 3 family models that are feasible to run on four H100 GPUs: LLAMA 3.2 1B, LLAMA 3.2 3B, and LLAMA 3.1 8B.

We evaluate training throughput and peak memory usage of three LONGSHIELD variants to understand the effect of CP context scaling, sharding and overlapped communication, and activation checkpointing.

- LONGSHIELD-V1 is a basic context-parallel implementation. It leverages existing efficiency tricks in the literature, except that it replaces the mixed ghost norm with the pure grad sample method to enable long contexts (cf. Section 2).

- LONGSHIELD-V2 builds on LONGSHIELD-V1 by (i) adding per-sample gradient sharding to save memory and (ii) overlapping the RS aggregation of per-sample gradients to boost throughput.

- Finally, LONGSHIELD-V3 applies DP-compatible full activation checkpointing at the transformer block level to LONGSHIELD-V2. LONGSHIELD-V3 further scales sequence length because full activation is not materialized and recomputed during the backward pass.

For a fair comparison, we implement ZeRO-DP+, an improved version of ZeRO-DP with enhanced memory efficiency and utility, within the same framework. We utilize recent FSDP ghost clipping (FGC) support from Opacus (Yousefpour et al., 2021), which provides FSDP2 (PyTorch Documentation, 2025) support and enables flat clipping for improved convergence (Bu et al., 2023b). We also incorporate all efficiency techniques developed in ZeRO-DP (Bu et al., 2023a), including mixed ghost norm (Bu et al., 2022), book-keeping (Bu et al., 2023d), and the origin parameter trick (Bu et al., 2023d).

### 4.2 ZERO-DP+ PERFORMANCE

We compare training throughput (TPS) at the maximum achieved sequence length (T) across various micro-batch sizes (MBS) for LLAMA 3.2 1B, LLAMA 3.2 3B, and LLAMA 3.1 8B over 4× H100 in Table 3. The majority of the runs incur non-trivial throughput overheads ranging from 15% to 156%.

| MBS | | 1B | | | | 3B | | | | 8B | | |
|-----|-----|--------|-------|------|-----|--------|-------|------|-----|--------|-------|------|
| | T | non-DP | DP | Gap | T | non-DP | DP | Gap | T | non-DP | DP | Gap |
| 1 | 16k | 28.8k | 25.0k | 1.15 | 8k | 14.2k | 11.3k | 1.26 | 4k | 7.18k | 4.30k | 1.67 |
| 2 | 8k | 35.7k | 24.2k | 1.48 | 4k | 15.7k | 11.8k | 1.33 | 2k | 7.64k | 2.98k | 2.56 |
| 4 | 4k | 40.8k | 29.1k | 1.40 | 2k | 16.8k | 7.22k | 2.33 | 1k | 7.79k | 3.90k | 2.0 |
| 8 | 2k | 44.1k | 22.6k | 1.95 | 1k | 17.9k | 9.70k | 1.84 | 512 | 7.91k | 4.47k | 1.77 |

Table 3: Training throughput (measured in TPS) comparison between non-DP and ZeRO-DP+ at the maximum achieved sequence length (power of 2) under various MBS for LLAMA 3.2 1B, LLAMA 3.2 3B, and LLAMA 3.1 8B on 4× H100 GPUs.

This slowdown is attributable to various sources, mostly notably due to ghost norm overheads, unoptimized communication for synchronizing private gradients, as well as DP optimizer overheads due to ineffective batching.

### 4.3 LONGSHIELD PERFORMANCE

We show training throughput and peak memory at the maximum achieved sequence length across various MBS for LLAMA 3.2 1B, LLAMA 3.2 3B, and LLAMA 3.1 8B over four H100 GPUs in Table 4. Our recommended settings run with MBS less than the number of GPUs (e.g., first two rows), as we use CP for extended context. The third row, where MBS equals the number of GPUs, is *not* recommended as it provides no context scaling compared to an FSDP setting; we include it for sensitivity analysis to help explain performance trends. In practice, one should avoid 4-GPU CP with MBS=4, and instead choose 4-GPU FSDP with MBS=1, which has the same achieved context but contexts are local to each GPU to avoid KV exchange under ring-attention Liu et al. (2023a).

(a) LLAMA 3.2 1B Throughput in TPS

| MBS | T | non-DP | ZeRO-DP+ | V1 | V2 | V1 gap | V2 gap |
|---|---|---|---|---|---|---|---|
| 1 | 64k | 11.8k | OOM | 11.4k | 11.6k | 1.04 | 1.02 |
| 2 | 32k | 16.4k | OOM | 15.1k | 15.8k | 1.09 | 1.04 |
| 4 | 16k | 20.9k | OOM | 18.1k | 19.4k | 1.15 | 1.08 |

(b) LLAMA 3.2 1B Peak Memory in GB

| non-DP | V1 | V2 | V1 gap | V2 gap |
|---|---|---|---|---|
| 69.0 | 70.6 | 66.2 | 1.02 | 0.96 |
| 68.4 | 70.5 | 66.7 | 1.03 | 0.98 |
| 68.0 | 71.2 | 66.4 | 1.05 | 0.98 |

(c) LLAMA 3.2 3B Throughput in TPS

| MBS | T | non-DP | ZeRO-DP+ | V1 | V2 | V1 gap | V2 gap |
|---|---|---|---|---|---|---|---|
| 1 | 32k | 6.47k | OOM | 5.95k | 6.08k | 1.09 | 1.06 |
| 2 | 16k | 8.31k | OOM | 7.02k | 7.42k | 1.18 | 1.12 |
| 4 | 8k | 9.34k | OOM | 5.87k | 7.91k | 1.59 | 1.18 |

(d) LLAMA 3.2 3B Peak Memory in GB

| non-DP | V1 | V2 | V1 gap | V2 gap |
|---|---|---|---|---|
| 67.0 | 75.1 | 66.2 | 1.12 | 0.99 |
| 67.0 | 75.1 | 66.0 | 1.12 | 0.99 |
| 66.9 | 76.9 | 66.0 | 1.15 | 0.99 |

(e) LLAMA 3.1 8B Throughput in TPS

| MBS | T | non-DP | ZeRO-DP+ | V1 | V2 | V1 gap | V2 gap |
|---|---|---|---|---|---|---|---|
| 1 | 16k | 4.44k | OOM | OOM | 4.08k | N/A | 1.09 |
| 2 | 8k | 4.82k | OOM | OOM | 4.09k | N/A | 1.18 |
| 4 | 4k | 4.99k | OOM | OOM | 2.20k | N/A | 2.27 |

(f) LLAMA 3.1 8B Peak Memory in GB

| non-DP | V1 | V2 | V1 gap | V2 gap |
|---|---|---|---|---|
| 73.2 | OOM | 72.2 | N/A | 0.99 |
| 73.3 | OOM | 76.4 | N/A | 1.04 |
| 73.3 | OOM | 76.6 | N/A | 1.05 |

Table 4: Throughput and peak memory at the maximum sequence length reached under CP

**LONGSHIELD context scaling capability.** Both LONGSHIELD-V1 and LONGSHIELD-V2 offer 4× context scaling over LLAMA 3.2 1B, LLAMA 3.2 3B, and LLAMA 3.1 8B, which is linear to the context-parallel degree(c.f. Table 4 and Table 3. ZeRO-DP (Bu et al., 2023a) gets OOM under the 4× H100 setting with LONGSHIELD's context length, and will get OOM even with **infinite** H100, due to hard single GPU activation memory ceiling.

LONGSHIELD significantly reduces the throughput gap between non-DP and DP (c.f. Table 4 and Table 3) as the proportion of DP overhead grows slower than attention. The absolute throughput of CP cannot beat that of FSDP (both non-DP and DP) due to longer context as well as CP framework overheads. But this is a justifiable compromise for emerging DP context extension CPT tasks, which typically require training fewer than 5B tokens (Fu et al., 2024) (compared to DP pretraining that requires trillions of tokens).

**Sharding and communication overlapping.** We compare performance of LONGSHIELD-V1 and LONGSHIELD-V2 to understand the effectiveness of sharding and communication. LONGSHIELD-V2 consistently outperforms LONGSHIELD-V1 in peak memory usage and in some cases even beats the non-DP baseline. This is because all our runs have GA enabled, as DP requires larger GBS. The non-DP baseline (TorchTitan) makes the design choice not to shard gradients until the end of the GA to avoid premature gradient sharding and unnecessary communication. LONGSHIELD-V2 instead always aggregates and shards per-sample gradients as we need to clip and accumulate per-sample gradients to free space between multiple forward and backward passes.

Although LONGSHIELD-V2 does pay additional communication compared with non-DP, our throughput impact is negligible as most of the communication can be overlapped. Communication can become a bottleneck for large MBS or large models (e.g. LONGSHIELD-V1 under LLAMA 3.2 3B with MBS=4 and even communication optimized LONGSHIELD-V2 under LLAMA 3.1 8B with MBS=4). However, they are the non-recommended, ill-formed CP scenarios that serve to enhance sensitivity understanding. For recommended settings, LONGSHIELD-V2 consistently cuts the throughput gap

between DP and non-DP by one third to a half. In general, intra-node GPUs benefit from the large communication bandwidth provided by NVLink. We expect the overlapping to play a more critical role when scaling beyond a single node, as inter-node interconnect bandwidth is limited.

**Activation checkpointing.**    We evaluate the performance of DP-aware activation checkpointing in Table 5. We list the throughput in TPS per GPU and the peak memory for non-DP (with and without AC), as well as LongShield-V2 (without AC) and LongShield-V3 (with AC).

| (a) Llama 3.2 1B Throughput in TPS | | | | | | (b) Llama 3.2 1B Peak Memory in GB | | | |
|---|---|---|---|---|---|---|---|---|---|
| MBS | T | non-DP | non-DP AC | V2 | V3 | non-DP | non-DP AC | V2 | V3 |
| 1 | 64k | 11.8k | 9.29k | 11.6k | 9.16k | 69.0 | 42.7 | 66.2 | 48.3 |
| 1 | 128k | OOM | 5.92k | OOM | 5.79k | OOM | 61.9 | OOM | 75.3 |
| 1 | 256k | OOM | OOM | OOM | OOM | OOM | OOM | OOM | OOM |

| (c) Llama 3.2 3B Throughput in TPS | | | | | | (d) Llama 3.2 3B Peak Memory in GB | | | |
|---|---|---|---|---|---|---|---|---|---|
| MBS | T | non-DP | non-DP AC | V2 | V3 | non-DP | non-DP AC | V2 | V3 |
| 1 | 32k | 6.47k | 5.26k | 6.08k | 4.99k | 67.0 | 35.9 | 66.2 | 38.8 |
| 1 | 64k | OOM | 3.84k | OOM | 3.71k | OOM | 58.1 | OOM | 62.7 |
| 1 | 128k | OOM | 2.45k | OOM | 2.31k | OOM | 76.7 | OOM | 76.9 |

| (e) Llama 3.1 8B Throughput in TPS | | | | | | (f) Llama 3.1 8B Peak Memory in GB | | | |
|---|---|---|---|---|---|---|---|---|---|
| MBS | T | non-DP | non-DP AC | V2 | V3 | non-DP | non-DP AC | V2 | V3 |
| 1 | 16k | 4.44k | 3.49k | 4.08k | 3.32k | 73.2 | 50.4 | 72.2 | 52.5 |
| 1 | 32k | OOM | 3.15k | OOM | 3.01k | OOM | 57.4 | OOM | 62.3 |
| 1 | 64k | OOM | 2.32k | OOM | 2.26k | OOM | 73.4 | OOM | 75.5 |

Table 5: Performance effect of Activation Checkpointing

LongShield-V3 achieves 2×, 4×, and 4× additional sequence scaling beyond LongShield-V2's maximum achieved sequence length, same amount of sequence scaling compared to the non-DP case. When comparing the same sequence length, AC has an expected slowdown (roughly 33% for additional forward) but with much smaller peak memory. The relative throughput gap further shrinks as the attention cost scales quadratically and dominates the runtime. In contrast, DP computation based on the grad sample methods avoids ghost norm and the complexity only scales linearly with sequence length.

LongShield-V3 memory usage can differ substantially from the non-DP AC primarily due to large fragmentation in the current Opacus implementation (Yousefpour et al., 2021). For example, the Llama 3.2 1B 128k sequence length run in Table 5b has identical 58.9 GB maximum active memory, but the fragmentation causes a huge difference between maximum reserved space (61.9 GB vs 75.3 GB). We leave better engineering optimization of Opacus (Yousefpour et al., 2021) as future work, as it does not affect our context-scaling results.

# 5 Conclusion

We introduce LongShield, a memory- and communication-efficient context-parallel DP training method that closes the performance gap to non-DP while enabling long-context scaling on modest GPU budgets. LongShield keeps per-sample gradients shards local to each GPU to avoid full materialization, overlaps per-sample gradient aggregation with backward computation to sustain throughput, and enables DP-safe activation checkpointing to extend context further. On Llama 3.1 8B with 4× NVIDIA H100 GPUs, LongShield scales sequence length from 4k to 16k compared to the state-of-the-art ZeRO-DP, achieves linear sequence-length scaling, shrinks the throughput gap from 67% to 8.9% while matching non-DP memory usage, and reaches a 64k context length with activation checkpointing. These results show that long-context DP training is practical on modest GPU budgets.

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

---

**Algorithm 1** Differentially Private SGD (DP-SGD)

---

**Require:** Dataset $D$; iterations $K$; batch size $B$; sampling rate $q = B/|D|$; clipping threshold $C$; noise multiplier $\sigma$; learning rates $\{\eta_k\}_{k=1}^K$.
 1: Initialize parameters $\theta_0$.
 2: **for** $k = 1$ to $K$ **do**
 3:     Sample a minibatch $B_k \subset D$ from a Poisson distribution with rate $q$.
 4:     **for all** $x \in B_k$ **do**
 5:         Compute per-example gradient $g_x \leftarrow \nabla_\theta \ell(\theta_{k-1}; x)$.
 6:         Clip: $\tilde{g}_x \leftarrow g_x \cdot \min\left(1, \frac{C}{\|g_x\|_2}\right)$.
 7:     **end for**
 8:     Aggregate clipped gradient: $\bar{g}_k \leftarrow \frac{1}{B}\left(\sum_{x \in B_k} \tilde{g}_x + \mathcal{N}(0, \sigma^2 C^2 \mathbf{I})\right)$.
 9:     Update parameters: $\theta_k \leftarrow \theta_{k-1} - \eta_k \bar{g}_k$.
10: **end for**
11: **return** $\theta_K$.

---

# A  PRELIMINARY AND RELATED WORK

**Differential privacy.** We adopt the standard $(\varepsilon, \delta)$-differential privacy (DP) definition (Dwork et al., 2014). Two datasets $D, D'$ are *neighbors* if they differ in a single individual record.

**Definition 1** ($(\varepsilon, \delta)$-DP). A randomized mechanism $\mathcal{M} : \mathcal{D} \to \mathcal{R}$ is $(\varepsilon, \delta)$-differentially private if for all measurable $S \subseteq \mathcal{R}$ and all neighboring $D, D'$,

$$\Pr[\mathcal{M}(D) \in S] \leq e^\varepsilon \Pr[\mathcal{M}(D') \in S] + \delta.$$

**DP-SGD.** DP-SGD (Abadi et al., 2016) privatizes stochastic gradient descent by bounding per-example sensitivity via gradient clipping and injecting calibrated Gaussian noise into the aggregated (mini-batch) gradient. Let $f_\theta$ denote the model with parameters $\theta$, loss $\ell(\theta; x)$ on example $x$, sampling rate $q = B/|D|$, clipping threshold $C > 0$, learning rate $\eta_t$, and noise multiplier $\sigma > 0$. DP-SGD is shown in Algorithm 1.

**Privacy accounting.** Across $K$ iterations, the overall $(\varepsilon, \delta)$ depends on the subsampling rate $q$, noise multiplier $\sigma$, and the number of steps $K$. Tight analyses typically use the *moments accountant* (Abadi et al., 2016) or *Rényi DP* (RDP) composition (Mironov, 2017), then convert back to $(\varepsilon, \delta)$ for a target $\delta$ (e.g., $\delta \approx 1/|D|$).

**Efficient and Scalable DP methods.** A major bottleneck in DP is evaluating the per-sample norm for clipping. Recent systems work aims to make private training approach the speed and memory profile of standard training, but is mostly effective for small contexts Li et al. (2021); Bu et al. (2023d). Yousefpour et al. (2021) evaluate per-sample norm by instantiating per-sample gradient over the entire network, adding heavy memory pressure. Li et al. (2021) introduces ghost clipping to calculate ghost norm and avoid instantiating per-sample gradients. However, it suffers a throughput penalty due to the required second backpropagation. Bu et al. (2023d) introduces a *book-keeping* (BK) technique that stages activation gradients to avoid redundant data gradients during second backpropagation. Complementary to single-node efficiency, SOTA distributed DP framework ZeRO-DP (Bu et al., 2023a) scales SOTA efficient DP methods Bu et al. (2022; 2023d) using zero redundancy optimizer (ZeRO) (Rajbhandari et al., 2020). However, two big issues remain. First, efficient DP methods are not fully supported over ZeRO-2/3 due to the engineering complexity of integrating into DeepSpeed (Rasley et al., 2020). Second, ZeRO-DP only supports per-module or per-parameter clipping. This simplifies the computational problem but at the cost of provably worse convergence (Bu et al., 2023b).

**Clipping styles.** Gradient clipping controls sensitivity in DP-SGD. The *flat/global* (Abadi et al., 2016; Yousefpour et al., 2021; Li et al., 2021; Bu et al., 2023c; Yu et al., 2021) variant clips the concatenated gradient with a single bound $C$, which is simple and often delivers better accuracy but incurs more system overhead (Either incur 67% throughput reduction due to a second backward pass (Lee & Kifer, 2020; Li et al., 2021; Bu et al., 2022), or requires additional memory penalty to

book-keep activation gradients (Bu et al., 2023d) or per-sample gradients (Yousefpour et al., 2021)). *Per-layer* (He et al., 2022) clipping instead enforces bounds $\{C_\ell\}$ per layer. It does not encounter performance overhead, since the layer gradient can be clipped immediately. However, it raises utility concerns where it has provably worse convergence (Bu et al., 2023b).

