# OpenReview forum: "LONGSHIELD: SCALABLE DISTRIBUTED DIFFERENTIALLY PRIVATE TRAINING FOR LONG-CONTEXT LLMS"
_ICLR.cc/2026/Conference — Submitted to ICLR 2026_

### Official Review · Reviewer_fsdT · 2025-10-19

**Soundness:** 2
**Presentation:** 1
**Contribution:** 2
**Rating:** 2
**Confidence:** 4

**Summary:**

This paper introduces LongShield which integrates context parallelism to DP (differentially private) training method aka DP-SGD. The main contribution of the paper is to integrate context parallelism on top of FSDP (ZERO-DP+) for training LLMs with large context using differentially private algorithms such as DP-SGD. The paper also enables DP-safe activation checkpointing to extend context further.  On Llama 3.1 8B with 4× NVIDIA H100 GPUs, LongShield scales sequence length from 4k to 16k compared to the state-of-the-art ZeRO-DP, achieves linear sequence-length scaling, shrinks the throughput gap from 67% to 8.9% while matching non-DP memory usage, and
reaches a 64k context length with activation checkpointing.

**Strengths:**

1. The paper identifies that for large context lengths, ghost clipping introduces significant memory overhead and shits to computing pure grad sample to integrate context parallelism to ZERP-DP+
2. LongShield keeps per-sample gradients shards local to each GPU to avoid full materialization, overlaps per-sample gradient aggregation with backward computation to sustain throughput.
3. The paper enables DP-safe activation checkpointing to extend context further.
4. The paper presents experimental results on Llama 3.2 1B, Llama 3.2 3B, and Llama 3.1 8B over 4× H100 GPU and show that LongShield scales sequence length from 4k to 16k compared to the state-of-the-art ZeRO-DP.

**Weaknesses:**

1. The main weakness of the paper is that LongShield computes per-gradient sample which is notorious for preserving the per-sample gradient over the entire model instead of using Fast Gradient Clipping (FGC) to avoid ghost clipping's $T^2$ overhead. This choice has not been justified if there is indeed some benefit (in-terms of throughput) in computing per-sample gradient of the entire model instead of FGC with two backward passes.
2. The paper was hard to follow, please add a pseudo-code similar to algorithm 1 for LongShield, ZERO-DP and ZERO-DP+. Please explain technical details of CP and how it integrates along with FSDP for DP-SGD more clearly.
3. No codebase is provided.

**Questions:**

Questions:
1. "We adopt the pure gradient-sample (GS) approach to avoid ghost overhead." -- This is unclear to me. I understand that Ghost Clipping introduces memory overhead proportional to $T^2$ but this can be eradicated by shifting to Fast Gradient Clipping. Why do we need to store the entire per-sample gradient instead of using Fasting Gradient Clipping + Two backwards passes if memory proportional to $T^2$ was indeed the bottleneck?
2. ZERO-DP works with per-layer gradient clipping which doesn't yield good utility in terms of test accuracy. For ZeRO-DP+ results, are the experiments conducted for flat clipping instead of per-layer?
3.  In my opinion, integrating CP and FSDP into DP-SGD with Fast Gradient Clipping is trivial. If this is not the case, then can the authors explain why?
4. "For example, mixed ghost norm choose ghost for Llama 3.1 8B final linear layer up to T= 16k. However, the ghost norm is 4× more FLOPs than directly evaluating the per-sample gradient, and the final dot product between two large intermediate tensors (𝑂(𝐵𝑇2)) causes a similar time, according to our profiling, due to the reduction nature." -- this statement is not clear to me. Mixed GC will pick GC or FGC based on minimum overhead condition and GC has more FLOPs than FGC because 1) FGC includes one matmul of BxTxp and BxTxd, where as 2) GC has two matmuls (BxTxp with BxpxT, BxTxd with BxdxT) and one dot product ($B \times T^2$ with $B \times T^2$). The statement says that GC has similar time according to the profile but causes 8x slowdown, this part is not clear.
5. Table 4 shows the results for ZERO-DP+ when it OOMs. How does LongShield perform in terms of throughput and peak memory when ZERO-DP+ doesn't OOM. Can we utilize LongShield to improve throughput even when ZERO-DP+ doesn't OOM?


Suggestions
1. In figure 1, mention the per-device batch-size and global batch-size. For example, in this particular figure, FSDP supports GBS of 2*MBS and a per-device batch-size of MBS. But with CP, we can support larger context length at the cost of per-device batch-size reduced to 1/2 and GBS of 1. This is not clear in the figure or in the paragraph below.
2. Figure 4 and text below is not consistent. The text says "we can all-to-all (A2A) exchange the activation tensor (shape changing from (MBS, T/2, p) to (MBS, T, p/2)) and then all-gather (AG) the activation gradient tensor (shape transferred from (MBS, T/2, d) to (MBS, T, d))." but the figure shows (MBS, T, p) and (MBS, T, d/2).  Please correct it.

---

> ### Author Response · Authors · 2025-12-03
>
> We thank the reviewer for their detailed feedback and for acknowledging that LongShield effectively identifies the memory bottlenecks of ghost clipping at large context lengths. We address your concerns regarding the choice of gradient computation method and clarify the system design below.
>
> **Choice of Gradient Computation: Per-Sample Gradient vs. Fast Gradient Clipping (Weakness 1, Question 1 & 3):**
>
> The reviewer asks why we compute full per-sample gradients (GS) instead of using Fast Gradient Clipping (FGC) / Ghost Clipping, suggesting FGC avoids memory overhead. We explicitly chose the GS approach for two critical reasons:
> 1. FGC inherently requires a second backward pass to calculate norms without materializing gradients. This imposes a strict theoretical limit on training speed, typically capping throughput at ~60% of the non-DP baseline. In contrast, LongShield achieves a throughput gap of only ~8.9% compared to the non-DP baseline. Therefore, adopting FGC would significantly increase the performance gap.
> 2. The reviewer is correct that GS is notorious for memory usage if unsharded. However, LongShield’s core contribution is the sharding of these per-sample gradients across the Context Parallel (CP) dimension. By sharding, we reduce the memory footprint to manageable levels without incurring the massive throughput penalty of FGC.
>
> **ZeRO-DP+ Clipping Configuration (Question 2):**
>
> For all ZeRO-DP+ experiments, we implemented flat clipping to ensure a fair comparison with equivalent utility to LongShield.
>
> **Documentation and Code (Weakness 2 & 3, Suggestions):**
>
> We will add formal pseudo-code for LongShield, ZeRO-DP, and ZeRO-DP+ in the Appendix of the final version to clarify the integration details.
>
> Yes, we are committed to releasing the full codebase to ensure reproducibility and facilitate further research.
>
> We appreciate the detailed eye. We will update Figure 1 to explicitly state per-device vs. global batch sizes and correct the tensor shape labels in Figure 4 to match the text description.
>
> **Clarification on Ghost Norm Overhead (Question 4):**
>
> In Mixed Ghost Clipping, selecting between Ghost Clipping and Fast Gradient Clipping is fundamentally about minimizing memory in the per-sample gradient computation. Ghost clipping can have 4× more FLOPs yet produce varied wall-time because reductions, memory traffic, and cross-device communication can dominate runtime. For the quoted statement, it is the combination of extra FLOPs and the time spent due to the reduction that causes the 8× slowdown. We have modified the statement in the paper to clarify what it means.
>
> **Performance when ZeRO-DP+ does not OOM (Question 5):**
>
> LongShield is specifically designed for the regime where ZeRO-DP+ fails (long context). In very short contexts where ZeRO-DP+ fits in memory, ZeRO-DP+ may have a slight advantage because it avoids the inter-GPU communication required by Context Parallelism. However, our communication overlap optimizations (LongShield-V2) minimize this gap. Table 4 shows this trend, where the performance gap increases as context length is reduced. The primary value of LongShield is enabling training in the 16k–64k context range, where ZeRO-DP+ simply cannot run. In this range, the performance gap from the non-DP baseline is small, making DP training more accessible under similar hardware availability.

---

### Official Review · Reviewer_vekz · 2025-10-28

**Soundness:** 3
**Presentation:** 2
**Contribution:** 3
**Rating:** 6
**Confidence:** 4

**Summary:**

DP-SGD is the most used algorithm for training ML models with differential privacy. Prior literature (DP-ZeRO) has scaled DP-SGD training to very large models, but fails to scale to longer sequence lengths. Through a series of optimizations this paper scales from max sequence length of 4k (prior work) to 16k on the Llama-3 8B model. They use optimizations from the exisiting literature on LLM scaling such as  (1) context parallelism, (3) gradient sharding (3) activation checkpointing, and adapt these to work with DP-SGD.

**Strengths:**

- The paper achieves significant results in terms of scaling DP-SGD training to longer contexts.
- Experimental results are comprehensive (although a bit hard to compare between methods given the separate tables for each method).
- The paper provides important contributions in adapting popular scaling techniques such as context parallel and activation checkpointing to DP-SGD training, thus enabling further research in private LLM training.
- I liked the insight on the limitations of ghost clipping for scaling to longer contexts.

**Weaknesses:**

- The contributions are mainly engineering-oriented versus conceptual/algorithmic since existing methods from the non-private literature are extended somewhat straightforwardly to the private case.

- The paper could be better self-contained. Several concepts are used without much explanation such as ghost clipping/ghost overhead, FSDP, context extension continue pre-training, the overlap of communication and computation in the input-stationary pattern.

**Questions:**

- Will there be open-source code for this paper? That would be very important given that a lot of the contributions are engineering-oriented and would enable ongoing research in this area.


More minor comments:
- What is the meaning of "large fragmentation" in in the current Opacus imlementation?
- Another relevant work/baseline might be "Scaling Private Deep Learning with Opacus: Advances for Large Language Models" which discuss FSDP with the ghost clipping approach.

---

> ### Author Response · Authors · 2025-12-03
>
> We thank the reviewer for the positive assessment and for highlighting the significant results we achieved in scaling DP-SGD to longer contexts. We are glad you appreciated the insights regarding the limitations of ghost clipping and the practical value of our adaptations for the research community. We address your specific questions and feedback below.
>
> **Engineering vs. Conceptual Novelty (Weakness 1):**
>
> We address the concern regarding "straightforward extensions" in our **General Response**. We demonstrate that a straightforward application of existing non-private scaling techniques to DP is insufficient due to the massive footprint of unsharded per-sample gradients. LongShield implements the necessary sharding strategy for extending context length that is absent from its current frameworks.
>
> Our contribution lies in (i) identifying the input-stationary communication pattern as the necessary architectural choice to solve the memory bottleneck and (ii) designing a novel hook management strategy that resolves the fundamental conflict between Opacus-style DP and Activation Checkpointing. We believe enabling a capability that was previously impossible (scaling from 4k to 64k context) is a significant research contribution.
>
> **Presentation and Definitions (Weakness 2):**
>
> We appreciate the feedback regarding the explanation of terms. In the final version, we will expand the "Preliminaries" section and Appendix to provide self-contained definitions of Ghost Clipping, FSDP (Fully Sharded Data Parallelism), and the specific mechanics of Context Extension/Continued Pre-training. We will ensure the trade-offs between these systems are clearly articulated for readers less familiar with the specific distributed training literature.
>
> **Open Source Code (Question 1):**
>
> Yes. We are fully committed to open-sourcing the LongShield code. We intend to release the repository at publication time.
>
> **Minor Comments (Question 2, 3)**
>
> The "large fragmentation" mentioned refers to CUDA memory fragmentation within the TorchTitan allocator.
>
> We are aware of the work "Scaling Private Deep Learning with Opacus" (Aketi et al., 2025). That work focuses on FSDP with ghost clipping. As we analyze in our paper, ghost clipping suffers from O(T^2) computational complexity and requires an All-Gather operation that becomes prohibitive at long sequence lengths. Our baseline comparison, DP-ZeRO, uses the exact same state-of-the-art realization of that FSDP+DP approach. This approach hits an OOM ceiling at 8k/16k tokens. LongShield deliberately moves away from ghost clipping to a pure gradient sample approach to avoid this quadratic bottleneck, which is why we can scale significantly further. We will explicitly cite and discuss this paper in our related work section to clarify this distinction in our final version.
>
> **References**
>
> Aketi, S.A., Bullock, W., Kalemaj, I., Ullah, E. and Zhang, H., 2025. Scaling Private Deep Learning with Opacus: Advances for Large Language Models. In Championing Open-source Development in ML Workshop@ ICML25.

---

### Official Review · Reviewer_2uzq · 2025-11-02

**Soundness:** 2
**Presentation:** 3
**Contribution:** 2
**Rating:** 2
**Confidence:** 4

**Summary:**

This paper proposes LongShield, a system approach that combines context parallelism (CP) with existing differentially private (DP) training frameworks (e.g., Opacus + ZeRO-DP) to enable long-context DP training for LLMs. The key contributions include per-sample gradient sharding, communication overlap, and DP-compatible activation checkpointing. Experiments on Llama-3 models (1B–8B) show improved throughput and reduced memory over ZeRO-DP, scaling context length up to 64k tokens on 4×H100 GPUs.

**Strengths:**

1. Clear and reproducible system engineering.

2. Demonstrates that DP-SGD can scale to longer contexts using modest hardware.

3. Addresses practical compatibility issues (e.g., checkpointing with DP hooks).

4. Experimental evaluation on real Llama-3 models is thorough for throughput and memory.

**Weaknesses:**

1. Weak novelty. The method essentially combines context parallelism with DP-SGD under existing frameworks. It does not introduce new algorithms, optimizers, or privacy accounting techniques.

2. Lack of multi-dimensional parallelism. The system is confined to single-node CP. It does not explore or support other critical dimensions of LLM parallelism — such as Tensor Parallel (TP), Pipeline Parallel (PP), Expert Parallel (EP), or ZeRO data sharding. The paper even notes communication challenges would “become more critical” beyond one node, but never verifies cross-node scalability. Consequently, the proposed method’s scalability under realistic distributed setups remains unclear.

3. No integration with established frameworks. CP is already well-implemented in Megatron, which also provides integrated TP/PP/EP interfaces. The authors should compare their CP-DP implementation against a baseline that simply adds DP-SGD into Megatron. Moreover, they should explain why they did not directly integrate into Megatron — doing so would have allowed evaluation under real multi-dimensional parallelism (ZeRO + TP + PP + CP + EP) and would strengthen the engineering contribution.

4. Unconvincing motivation. The paper claims that DP is critical for long-context LLMs because “long sequences may contain sensitive data,” but provides no empirical evidence (e.g., no memorization or membership-inference study) to support that assumption.

5. No privacy evaluation. The paper lacks ε/δ reporting, attack-resistance analysis, or privacy–utility trade-off curves. As such, it demonstrates feasibility, not utility.

6. Limited scientific insight. Improvements (gradient sharding, hook fix, communication overlap) are incremental engineering optimizations that could be implemented in existing systems with minor effort.

**Questions:**

1. What privacy budgets (ε, δ) were achieved in your experiments?

2. How does DP noise affect model accuracy or perplexity?

3. Have you compared your CP implementation with Megatron’s sequence-parallel engine?

4. Could LongShield be integrated with Megatron to combine TP/PP/CP/DP?

5. How does performance scale across nodes with slower interconnects (e.g., RDMA instead of NVLink)?

---

> ### Author Response · Authors · 2025-12-03
>
> We thank the reviewer for the detailed assessment and for recognizing our system engineering, Llama-3 evaluation, and the practical value of enabling DP training on modest hardware. We address your concerns about novelty, framework choice, and evaluation below.
>
> **Novelty and Scientific Insight (Weakness 1, 6):**
>
> We respectfully clarify that our contribution is **identifying and solving system bottlenecks that prevent long-context DP training**, rather than proposing a new mathematical definition of privacy. Existing SOTA like ZeRO-DP (Bu et al., 2023) cannot scale to long sequences because they reach a single-GPU memory ceiling for unsharded activations under FSDP, even with infinite GPUs.
>
> **Motivation and Privacy Utility Trade-Offs (Weakness 4, 5  Question 1, 2):**
>
> We argue that DP is critical for long-context domains specifically because they involve **sensitive long-form data** such as full patient medical records and proprietary codebases. These data sources are key targets of inference attacks where DP is able to provide formal protection.
>
> We emphasize that LongShield imposes **no algorithm changes to the security guarantee of DP-SGD**. We use standard flat clipping, which is widely considered the strongest and most robust method for differential privacy (He et al., 2023). Because the underlying algorithm is unchanged, the privacy-utility trade-offs (epsilon/delta curves) are theoretically identical to standard DP-SGD; our contribution is making the computation feasible in memory while minimizing the memory overheads.
>
> We are currently finalizing the privacy evaluation to include in the final version. We will be testing the model’s long context capability through the Needle In A Haystack benchmark (Gregory, 2023). The specific privacy budgets (epsilon/delta) used in our experiments and the resulting convergence behavior will be reported in the final version.
>
> **Framework Selection (TorchTitan vs. Megatron) (Weakness 3 Question 3, 4):**
>
> We wanted to clarify that LongShield was built on top of TorchTitan (Liang et al., 2025), an established and peer-reviewed framework developed by the PyTorch Foundation community. All the required functionalities were well supported. We chose TorchTitan because it is a PyTorch-native solution that is easier to extend for research purposes than Megatron.
>
> **Multi-dimensional Parallelism (Weakness 2 Question 4):**
>
> Our experiments focused on Context Parallelism (CP) due to the resource constraints of our 4x H100 setup; however, the architectural principles of LongShield are compatible with the baseline Tensor Parallelism (TP) and Pipeline Parallelism (PP) implementations in the TorchTitan framework.
>
> **Scalability across nodes with slower interconnects (Weakness 2 Question 5):**
>
> We acknowledge that while CP requires communication, it is not a fundamental bottleneck like the memory capacity overheads in CP. This is because we optimized for communication by selecting an input-stationary communication pattern using Reduce-Scatter (RS). While NVLink is ideal, this overlap mechanism helps mitigate the slowdown on slower interconnects compared to standard All-Gather approaches.
>
> Scaling LongShield’s context parallelism across multiple nodes is likely ineffective due to communication bottlenecks. However, multi-dimensional parallelism using LongShield’s context parallelism within nodes and tensor or pipeline parallelism between nodes would be a viable strategy that still efficiently scales with context-length in a similar manner to the single node case. Although multi-node scalability for LongShield is not evaluated in this paper, multi-dimensional parallelism support including Tensor Parallel and Pipeline Parallelism, does exist within the TorchTitan framework and is compatible with the LongShield design.
>
> **References**
>
> Bu, Z., Chiu, J., Liu, R., Zha, S. and Karypis, G., 2023. Zero redundancy distributed learning with differential privacy. arXiv preprint arXiv:2311.11822.
>
> Gregory, K., 2023. Needle In A Haystack - pressure testing LLMs. Github.
>
> He, J., Li, X., Yu, D., Zhang, H., Kulkarni, J., Lee, Y.T., Backurs, A., Yu, N. and Bian, J., 2023. Exploring the Limits of Differentially Private Deep Learning with Group-wise Clipping. In The Eleventh International Conference on Learning Representations.
>
> Liang, W., Liu, T., Wright, L., Constable, W., Gu, A., Huang, C.C., Zhang, I., Feng, W., Huang, H., Wang, J. and Purandare, S., 2025. TorchTitan: One-stop PyTorch native solution for production ready LLM pretraining. In The Thirteenth International Conference on Learning Representations.

---

### Official Review · Reviewer_choR · 2025-11-03

**Soundness:** 3
**Presentation:** 3
**Contribution:** 2
**Rating:** 2
**Confidence:** 4

**Summary:**

This paper presents LongShield, a distributed framework for differentially private (DP) training of large language models (LLMs) under long-context settings. The method integrates context parallelism (CP) with per-sample gradient computation to achieve long-sequence scalability while maintaining DP guarantees.

**Strengths:**

1. Addresses an underexplored but important problem: long-context DP training.

2. Demonstrates strong empirical results with practical improvements in scalability and throughput.

3. Provides a clear implementation pathway compatible with Opacus and TorchTitan frameworks.

**Weaknesses:**

1. The core idea (computing per-sample gradients under CP) is a direct extension of standard CP, replacing the local (p, d) gradient computation with (B, p, d) per-sample gradients. There is no new parallelism algorithm or architectural innovation beyond this dimensional extension.

2. The “input-stationary vs. output-stationary” trade-off is not new; it mirrors prior analyses in FlashAttention-2 and Megatron-Ulysses. The overlap strategy is a routine engineering optimization rather than a conceptual advance.

3. The paper emphasizes that FSDP cannot shard per-sample gradients while CP can, but this follows trivially from the existing data partitioning dimensions. It is a property of CP’s tensor layout, not a new design.

4. The hook management fix for Opacus checkpointing is more of an implementation detail than a contribution; it does not introduce a new checkpointing strategy or fundamental compatibility solution.

5. There is no formal analysis of privacy-utility trade-offs or communication complexity, and most claims rely purely on empirical evidence.

6. The authors do not justify why long-context DP training is necessary or beneficial. While long-context modeling is relevant for general LLMs, it remains unclear whether the same motivation applies to private training. The experiments focus on throughput and context length scaling but omit any analysis of model utility, privacy-utility trade-offs, or downstream performance (e.g., perplexity, zero-shot tasks).

**Questions:**

1. Could the authors clarify whether any communication primitives or kernel fusion were re-implemented, or are all collectives reused from Megatron CP/Ulysses?

2. How does LongShield differ in actual code structure from integrating DP-SGD directly into existing CP frameworks?

3. Would privacy accounting or gradient clipping strategies change under model-parallel settings, or is the approach orthogonal to parallel dimension?

4. Why is long-context DP training practically needed? Are there real-world datasets or use cases that specifically require both privacy and long context?

---

> ### Author Response · Authors · 2025-12-03
>
> We thank the reviewer for recognizing that LongShield addresses an important, underexplored problem and for highlighting our strong empirical results and clear implementation pathway. We appreciate the opportunity to clarify our contributions regarding novelty, implementation, and motivation.
>
> **Novelty and Scientific Insight (Weakness 1, 2, 3, 4 Question 2):**
>
> We respectfully disagree with this characterization of the contribution as “trivial”. Our primary goal is to identify and enable long-context DP training, a task that is currently impossible with SOTA solutions like DP-ZeRO due to memory ceilings. This problem cannot be solved by simply "plugging in" existing CP implementations to DP frameworks. As detailed in our **General Response**, a direct integration of DP-SGD into CP frameworks does not effectively implement sharding for per-sample gradients.
>
> **Motivation and Privacy Utility Trade-Offs (Weakness 6  Question 4):**
>
> We argue that DP is critical for long-context domains specifically because they involve sensitive long-form data, such as full patient medical records and proprietary codebases. These are exactly the types of data that are prone to inference attacks, where DP provides formal protection. Existing DP pretraining is currently limited to very short contexts (Sinha et al., 2025) due to scaling issues, as we analyze in our paper. Short context lengths are insufficient for many modern applications involving long-form data. Using LongShield as a solution increases the achievable context length, making DP an accessible option for workloads operating on long-form data.
>
> **Privacy Accounting and Utility (Weakness 5 & Question 3):**
>
> We clarify that LongShield utilizes SOTA flat-clipping (De et al., 2022), which offers the **strongest privacy utility** (He et al., 2022). The parallel strategy is orthogonal to privacy accounting. Because LongShield introduces no changes to the underlying DP algorithm, the formal privacy analysis and utility curves are identical to standard DP-SGD.
>
> **Kernel Fusion (Question 1):**
>
> We did not implement new kernel fusion primitives because the **bottleneck for long-context DP is memory capacity, not compute-bound operations**. We utilized existing collectives but orchestrated them differently. Kernel optimization is a valid direction for future work, but it is not required to solve the immediate OOM blocking issue.
>
> **References**
>
> De, S., Berrada, L., Hayes, J., Smith, S.L. and Balle, B., 2022. Unlocking high-accuracy differentially private image classification through scale. arXiv preprint arXiv:2204.13650.
>
> He, J., Li, X., Yu, D., Zhang, H., Kulkarni, J., Lee, Y.T., Backurs, A., Yu, N. and Bian, J., 2023. Exploring the Limits of Differentially Private Deep Learning with Group-wise Clipping. In The Eleventh International Conference on Learning Representations.
>
> Sinha, A., Mesnard, T., McKenna, R., Liu, D., Choquette-Choo, C.A., Huang, Y., Yu, D., Kaissis, G., Charles, Z., Liu, R. and Chua, L., 2025. VaultGemma: A Differentially Private Gemma Model. arXiv preprint arXiv:2510.15001.

---

### Author Response · Authors · 2025-12-03
**Novelty: Architectural Necessity vs. Engineering Optimization (Why Naive Integration Fails)**

We write this general response to address a common question raised across reviews, reviewers asked whether LongShield is a "straightforward extension" of existing Context Parallelism (CP) to DP. We respectfully argue that a straightforward extension is quantifiably impossible on current hardware due to a unique memory bottleneck absent in non-private training.

To quantify this, we define the variables used in our analysis:

- $B$: Micro-Batch Size (MBS) per GPU.
- $p, d$: The output and input dimensions of a linear layer.
- $N$: The number of GPUs (Context Parallel size).

In standard backpropagation, gradients are summed over the batch, producing a tensor of size $(p \times d)$. However, DP training requires the per-sample gradient, which has the shape $(B, p, d)$. This tensor must be instantiated to compute norms for clipping.

Consider training Llama 3 8B with a micro-batch size of just $B=2$ using BF16 precision (2 bytes):
- Total Parameters ($P$): $\approx 8$ billion.
- Unsharded Per-Sample Gradient Size: $B \times P \times 2 \text{ bytes}$:
  - $$2 \times 8,000,000,000 \times 2 \text{ bytes} \approx \mathbf{32 \text{ GB}}$$

**Why Naive Integration Fails vs. LongShield:**

- Naive CP+DP: If one simply "plugs in" CP, the framework splits the activations ($T/N$), but the per-sample gradient calculation remains local to the operation. This forces each GPU to materialize the full 32 GB tensor in addition to the model weights (~16 GB), optimizer states, and the $O(T)$ activations required for long contexts. On an 80GB H100, this leaves almost zero room for the actual sequence tokens, capping context length at ~4k.
- LongShield: We implement specific logic to shard these per-sample gradients across the CP dimension immediately upon computation. With a CP size of $N=4$, the memory burden per GPU drops from 32 GB $\rightarrow$ 8 GB.

This 24 GB difference (per layer accumulation) is the decisive factor that allows us to reallocate memory to the sequence dimension, scaling context from 4k to 64k. This is not a trivial parameter change; it is a fundamental difference in memory layout required to make DP feasible.

---

### Meta-Review · Area_Chair_Bbea · 2026-01-10

**Summary:**

The reject decision is primarily driven by consensus regarding the submission's limited technical novelty and narrow experimental scope. The reviewers found the proposed framework a straightforward engineering extension of existing Context Parallelism (CP) techniques to Differentially Private (DP) training, specifically the dimensional expansion to per-sample gradients, rather than a fundamental algorithmic or architectural innovation. The evaluation was viewed as insufficient due to its restriction to single-node setups, with a notable lack of validation for multi-node scalability or integration with standard multi-dimensional parallelism strategies (such as Tensor or Pipeline Parallelism). Moreover, the absence of a rigorous privacy-utility analysis, including missing DP accounting and trade-off curves limits the paper's contribution relative to the current SOTA. In its present form, the paper does not meet the bar for publication at ICLR.

**Reviewer Concerns:**

The authors clarified some engineering and implementation details during the rebuttal phase. Specifically, they addressed a confusion regarding the choice of pure gradient-sample computation over ghost clipping. They also committed to open-sourcing the code, clarified the "large fragmentation" issue, and explained their choice of TorchTitan over Megatron. However, significant concerns regarding the fundamental contribution and evaluation remain outstanding.

**Reviewer Scores:**

It is unlikely that the consensus would have shifted significantly to support acceptance. Reviewers choR and 2uzq would likely have maintained their reject ratings, as the rebuttal focused on architectural justifications without providing the missing empirical evidence that was central to their critiques of the paper's novelty and scope. Reviewer fsdT might have marginally raised their score upon understanding the throughput advantages of the proposed method over ghost clipping, but their concerns regarding the paper's presentation would likely have prevented a shift to acceptance. Reviewer vekz would likely have maintained their marginally positive score, satisfied by the commitment to open-source the code and clarifications on memory fragmentation, but they would likely have remained the sole supporter of the work.

---

### Decision · Program_Chairs · 2026-01-26

Reject